# Integrated Comparative Transcriptome and circRNA-lncRNA-miRNA-mRNA ceRNA Regulatory Network Analyses Identify Molecular Mechanisms Associated with Intramuscular Fat Content in Beef Cattle

**DOI:** 10.3390/ani13162598

**Published:** 2023-08-11

**Authors:** Vahid Dehghanian Reyhan, Farzad Ghafouri, Mostafa Sadeghi, Seyed Reza Miraei-Ashtiani, John P. Kastelic, Herman W. Barkema, Masoud Shirali

**Affiliations:** 1Department of Animal Science, University College of Agriculture and Natural Resources, University of Tehran, Karaj 77871-31587, Iran; vahid.dehghaniya@ut.ac.ir (V.D.R.); farzad.ghafouri@ut.ac.ir (F.G.); ashtiani@ut.ac.ir (S.R.M.-A.); 2Faculty of Veterinary Medicine, University of Calgary, Calgary, AB T2N 4N1, Canada; jpkastel@ucalgary.ca (J.P.K.); barkema@ucalgary.ca (H.W.B.); 3Agri-Food and Biosciences Institute, Hillsborough BT26 6DR, UK; 4School of Biological Sciences, Queen’s University Belfast, Belfast BT9 5AJ, UK

**Keywords:** intramuscular fat content, ceRNA regulatory network, comparative transcriptome, meat quality, beef cattle

## Abstract

**Simple Summary:**

Meat quality and human-health-related indexes are important traits in beef cattle breeding. Intramuscular fat content (IMF) is a major meat quality trait that influences aroma, tenderness, and juiciness. Our objective was to integrate comparative transcriptomic and competing endogenous RNA (ceRNA) network analyses to identify candidate messenger RNAs (mRNAs) and regulatory RNAs involved in molecular regulation of longissimus dorsi muscle (LDM) tissue for IMF and fat metabolism of five beef cattle breeds (Angus, Chinese Simmental, Luxi, Nanyang, and Shandong Black). This study identified the primary metabolic-signaling pathways associated with IMF and fat metabolism, including calcium, cGMP-PKG, thyroid hormone, oxytocin signaling, and other metabolic pathways. In addition, genes *MCU*, *CYB5R1*, and *BAG3* were common among the 10 comparative groups that were involved in most of the terms related to fat storage and metabolic process. Perhaps differences in expression levels of lipid-metabolism-related RNAs demonstrated molecular factors underlying beef cattle breed differences in IMF and fat metabolism. The results of this study could inform marker-assisted selection and identify regulatory molecular mechanisms associated with IMF in beef cattle.

**Abstract:**

Intramuscular fat content (IMF), one of the most important carcass traits in beef cattle, is controlled by complex regulatory factors. At present, molecular mechanisms involved in regulating IMF and fat metabolism in beef cattle are not well understood. Our objective was to integrate comparative transcriptomic and competing endogenous RNA (ceRNA) network analyses to identify candidate messenger RNAs (mRNAs) and regulatory RNAs involved in molecular regulation of longissimus dorsi muscle (LDM) tissue for IMF and fat metabolism of 5 beef cattle breeds (Angus, Chinese Simmental, Luxi, Nanyang, and Shandong Black). In total, 34 circRNAs, 57 lncRNAs, 15 miRNAs, and 374 mRNAs were identified by integrating gene ontology (GO) and Kyoto Encyclopedia of Genes and Genomes (KEGG) enrichment analyses. Furthermore, 7 key subnets with 16 circRNAs, 43 lncRNAs, 7 miRNAs, and 237 mRNAs were detected through clustering analyses, whereas GO enrichment analysis of identified RNAs revealed 48, 13, and 28 significantly enriched GO terms related to IMF in biological process, molecular function, and cellular component categories, respectively. The main metabolic-signaling pathways associated with IMF and fat metabolism that were enriched included metabolic, calcium, cGMP-PKG, thyroid hormone, and oxytocin signaling pathways. Moreover, *MCU*, *CYB5R1*, and *BAG3* genes were common among the 10 comparative groups defined as important candidate marker genes for fat metabolism in beef cattle. Contributions of transcriptome profiles from various beef breeds and a competing endogenous RNA (ceRNA) regulatory network underlying phenotypic differences in IMF provided novel insights into molecular mechanisms associated with meat quality.

## 1. Introduction

Intramuscular fat (IMF), also known as marbling, has a major role in various meat quality traits. The composition of fatty acids in IMF, especially polyunsaturated fatty acids (PUFA), affects beef quality, with IMF closely related to various sensory characteristics, including taste, juiciness, and tenderness [1,2,3,4]. A critical aspect of beef fatty acid composition is the ratio of ω-6 and ω-3 fatty acid groups in PUFAs [4]. Differences in IMF and its fatty acid composition among breeds and genotypes of beef cattle are attributed to genetic factors [2,3] plus environmental factors (age, gender, nutrition, and management). The IMF content in the longissimus dorsi muscle (LDM) in Japanese black cattle is almost five-fold that of Angus and Holstein Friesian breeds [5]. In addition, IMF is related to fat metabolism (adipocyte number and size) and the balance between lipogenesis and lipolysis) [4].

As adipose tissue is critical in IMF and energy homeostasis, evaluating expression events of adipogenesis, a complex and coordinated process between coding and non-coding RNA molecules, could elucidate related cellular events [3]. Furthermore, key genes involved in fat metabolism could be genetic markers for improving carcass quality [4,6]. Evolving omics technologies, high-throughput datasets, and advanced computations facilitate the merging of complementary evidence from multiple levels of omics to create novel insights into complex regulatory mechanisms [6,7]. Comparisons of differential expression profiles in LDM tissue between beef cattle breeds with high- vs. low-IMF indicated that many genes involved in fat metabolism are related to IMF in beef cattle. These genes, which included *ADIPOQ* (Adiponectin, C1Q, Collagen Domain Containing), *FABP4* (Fatty Acid Binding Protein 4), *THRSP* (Thyroid Hormone Responsive), and *PPARG* (Peroxisome Proliferator-Activated Receptor Gamma), were identified as genetic markers related to fat metabolism in beef cattle [2,6,7,8,9]. Moreover, a genome-wide association study (GWAS) confirmed that IMF was affected by *PPARGC1A* (PPARG Coactivator 1 Alpha), *FOXP3* (Forkhead Box P3), and *HNF4G* (Hepatocyte Nuclear Factor 4 Gamma) [10].

Several genes involved in IMF and fat metabolism have been identified. However, the roles of non-coding RNAs (circRNAs, lncRNAs, and miRNAs) are not fully clarified. According to the ceRNA (competitive endogenous RNA) hypothesis, non-coding RNA regulatory species (e.g., circRNAs and lncRNAs) and protein-coding RNAs can act as competitive endogenous RNAs by binding to miRNA sites [11]. Integrating circRNA–lncRNA–miRNA–mRNA ceRNA networks provided insights into complex molecular mechanisms by considering various regulatory RNAs [12,13,14,15]. Long non-coding RNAs (lncRNAs) are single-stranded RNA molecules >200 nucleotides that regulate mRNA/gene expression at transcriptional, RNA processing, and translational process levels [16]. Recently, three lncRNAs affecting adipogenesis were discovered; lncFAM200B and NDUFC2AS stimulated differentiation of pre-adipocytes into mature adipocytes with positive regulatory effects, whereas ADNCR suppressed adipogenesis by targeting miR-204 [6,17]. Moreover, microRNAs (miRNAs) are non-coding RNA molecules containing 21 to 23 nucleotides involved in RNA silencing and post-transcriptional regulation of gene expression [18]. Two of these, miR-23a and miR-378 miRNAs, were involved in the growth and differentiation of muscle cells and the thickness of subcutaneous back fat [3]. CircRNAs acted as “molecular sponges” to absorb miRNAs and inhibit their regulatory actions, usually increasing expression of target genes. Furthermore, circFUT1O, circFGFR4, and circ-LMO7 circRNAs recruit miR-133a, miR-107, and miR-378a-3p miRNAs, respectively, and affect skeletal muscle differentiation [19,20].

Multi-part network approaches integrating various aspects of the transcriptome have provided novel insights regarding the regulation of complex, multigenic traits [15,21,22]. In this study, a comparative transcriptomic analysis of five beef cattle breeds (Angus, Chinese Simmental, Luxi, Nanyang, and Shandong Black), plus literature mining, were performed to identify related genes and lncRNAs, their functions, and important pathways. Furthermore, a circRNA–lncRNA–miRNA–mRNA ceRNA regulatory network was constructed, and its subnets were determined to better understand molecular mechanisms responsible for IMF and fat metabolism.

## 2. Materials and Methods

The overall design for data mining, preparation, and analysis of DE genes related to intramuscular fat (IMF) content in beef cattle is presented in Figure 1.

### 2.1. Data Collection

Initially, RNA-Seq datasets from the Gene Expression Omnibus (GEO) from the National Center for Biotechnology Information (NCBI) were used to identify candidate RNAs, protein–protein (PPI) interactions, construct a circRNA–lncRNA–miRNA–mRNA ceRNA regulatory network and its subnets, and determine metabolic and signaling pathways related to IMF and fatty acid composition in *Bos taurus*. GEO accession numbers for RNA-Seq samples from five beef cattle breeds are in Table 1. Shandong black cattle and Luxi cattle, in the GSE161272, were obtained from Shandong Black Cattle Technology Co., Ltd., and Dadi Luxi Cattle, respectively. The genetic basis of Shandong Black cattle is Japanese black, Luxi, and Bohai black cattle, with the breed approved by the National Animal and Poultry Genetic Resources Committee [7]. Luxi is a native beef cattle breed widespread throughout central China, with a reputation for meat freshness, tenderness, and flavor, plus high meat production capacity [7]. These cattle were fed roughage and concentrate (according to standard NY5127—2002 pollution-free feeding management of beef cattle) thrice daily with ad libitum access to drinking water and raised for 18 months at Qingdao Agricultural University in China. Six healthy and mature animals (3 Shandong black cattle and 3 Luxi cattle) were selected and slaughtered according to GBT19477-2004 cattle slaughtering procedures. The LDM (between ribs 12–13) was excised and cut into 2–3 cm^3^ cubes that were put into cryopreservation tubes and immersed in liquid nitrogen for transport to the laboratory for RNA extraction and sequencing [7]. GSE171876 included six samples of LDM of Angus (3 samples) and Chinese Simmental (3 samples), selected from Gansu Zhangye Qilian Muge Co., Ltd. (Zhangye, China). Angus cattle are renowned as a carcass breed, widely used in crossbreeding to improve meat quality and milking ability. Angus cattle in China are purebreds, recently imported from Scotland. Chinese Simmental cattle are a cross between Simmental from Switzerland and a local Chinese breed, with a history of 50 years; they are adapted for milking ability and meat production [23]. Angus and Chinese Simmental cattle 18 months of age were fed the same diet. After slaughter, 10 g of LDM tissue from each animal was collected and packaged into 2 mL sterile cryogenic vials and frozen in liquid nitrogen for RNA sequencing [23]. The GEO accessions of GSM4131022, GSM4131023, and GSM4131024 were selected from GSE139102 which was stored by Huang et al., 2020. Twelve Nanyang cattle were raised in the breeding center of Nanyang cattle (Nanyang, China). Nanyang cattle breed are important meat animals in China, with tender meat with high IMF but slow growth and low dressing percentage [24]. They were weaned at 3 months of age, castrated at 6 months, started to fatten at 18 months, and slaughtered at 30 months. Portions of LDM tissue were excised immediately after slaughter and frozen in liquid nitrogen for RNA extraction [2,25].

### 2.2. Differential Gene Expression Analysis

A quality check of raw fastq data was conducted with FastQC software (v0.11.9) [26]. Then, based on raw data quality control issues, noninformative sequences, as well as PCR primers and adapters, were trimmed using Trimmomatic software (v0.38) [27]. Mapping and sequence alignments were conducted on the *Bos taurus* reference genome (http://ftp.ensembl.org/pub/release-103/fasta/bos_taurus/dna/ (accessed on 18 February 2023)) using HISAT2 software (v2.2.1) [28]. FeatureCounts software (v2.0.3) was used to measure the total read counts of mapped sequences [29]. Finally, DESeq2 software (v2.11.40.7) was used to identify differentially expressed mRNAs and miRNAs of 10 pairwise comparisons: Angus vs. Chinese Simmental, Angus vs. Luxi, Angus vs. Nanyang, Angus vs. Shandong Black, Chinese Simmental vs. Luxi, Chinese Simmental vs. Nanyang, Chinese Simmental vs. Shandong Black, Luxi vs. Nanyang, Luxi vs. Shandong Black, and Nanyang vs. Shandong Black. For these, a threshold of a log fold change (FC) ≥1 and ≤−1 and a false discovery rate (FDR) ≤0.05 for significant DE mRNAs and miRNAs were obtained [30] (Appendix A). In addition, miRNAs in the RNA-Seq datasets were identified.

### 2.3. Literature Mining to Discover Relevant circRNAs, lncRNAs and miRNAs to IMF

Various online databases were examined to discover candidate circRNAs, lncRNAs, and miRNAs relevant to comprehensive literature mining. Online search databases and papers included Google Scholar, PubMed, Web of Science, and CrossRef from 2019 to 2023, with no language restrictions. Search terms consisted of both keywords and database-specific subject headings for the ceRNA regulatory network and IMF in beef cattle skeletal muscle tissue: breeds—beef cattle; practical tools—RNA-Seq; and outcome—ceRNA network or Regulatory RNAs—Intramuscular fat content trait. Keywords included beef cattle, intramuscular fat, longissimus dorsi muscle, lncRNA, circRNA, and ceRNA networks. First, identifiers and synonyms for each framework element were merged by applying the Boolean operator “OR”. Second, elements of the framework were merged by applying the Boolean operator “AND”. The identified non-coding RNAs (i.e., circRNAs, lncRNAs, and miRNAs) list was extracted as Appendix A.

### 2.4. Determining the Main RNAs List

To identify candidate RNAs (mRNAs and miRNAs) related to IMF content from RNA-Seq analysis, the number of common DE RNAs between eight comparative groups that were analyzed using the “Calculate and draw custom Venn diagrams” online tool was considered (https://bioinformatics.psb.ugent.be/webtools/Venn/ (accessed on 21 February 2023)) as shown in Appendix A. Subsequently, Appendix A (from RNA-Seq analysis) was integrated with the list of non-coding RNAs (Appendix A) from literature mining as the main RNAs list.

### 2.5. Functional Enrichment Analysis and KEGG Pathways

Gene ontology (GO) and enrichment analyses to explore the relevant metabolic pathways related to IMF and fat metabolism in beef cattle, including the biological process (BP), molecular function (MF), and cellular component (CC) of the main RNAs list (Appendix A), were performed using the online web tools DAVID (Database for Annotation, Visualization, and Integrated Discovery; [31]), PANTHER (Protein Analysis Through Evolutionary Relationships; [32]), GeneCards (www.genecards.org/ (accessed on 28 February 2023)), as well as the STRING database (https://string-db.org (accessed on 28 February 2023)) a comprehensive online web tool to define interactions between mRNAs/genes using a probabilistic confidence score and relevant pathways [33]. Enrichment of the signaling pathways relevant to identified mRNAs was provided in KEGG (Kyoto Encyclopedia of Genes and Genomes; https://www.genome.jp (accessed on 28 February 2023)); GO terms and pathways with FDR <0.05 were assumed significant and used to enrich the main gene list.

### 2.6. Identification of Regulatory RNAs and Target Gene Prediction

Potential target mRNAs were searched in the miRBase ([34]; https://www.mirbase.org/ (accessed on 2 March 2023)) and miRWalk (http://mirwalk.umm.uni-heidelberg.de/ (accessed on 2 March 2023)) databases. Targeted mRNAs were submitted to the DAVID and STRING databases to identify the enrichment target genes of each miRNA. Other targeted interactions between regulatory RNAs were predicted using: LNCipedia database ([35]; https://lncipedia.org (accessed on 2 March 2023)), NONCODE database ([36]; http://www.noncode.org/ (accessed on 2 March 2023)), and CircInteractome web tool ([37]; https://circinteractome.nia.nih.gov/ (accessed on 2 March 2023)).

### 2.7. Reconstruction of circRNA–lncRNA–miRNA–mRNA ceRNA Regulatory Network and Its Clustering Analysis

The circRNA–lncRNA–miRNA–mRNA ceRNA regulatory network was constructed based on circRNA–miRNA, circRNA–mRNA, lncRNA–mRNA, lncRNA–miRNA, and miRNA–mRNA interactions documented in related papers and online interaction databases. Protein–protein interaction (PPI) data extraction and gene regulatory networks (GRN) analyses were performed using the STRING database (https://string-db.org (accessed on 28 February 2023)), an online web tool that uses seven primary resources of interaction/association data including neighborhood, co-occurrence, fusion, experimental, co-expression, database, and literature mining to describe protein–protein interactions using a probable confidence score [33]. Cytoscape software is an offline tool with various plugins for analyses, including screening, integrating, and visualizing interactive data (v3.8.2) (National Institute of General Medical Sciences, Bethesda Softworks, Rockville, MD, USA; [38]. In the ceRNA regulatory network, biological molecule species (RNAs) and their interaction relationships were represented as nodes and edges, respectively. Utilized clustering on the circRNA–lncRNA–miRNA–mRNA ceRNA regulatory network was evaluated using MCODE [39], a Cytoscape plugin, to explore functional clusters and hub nodes. The MCODE plugin can be used to identify clusters for directed or undirected networks [39]. Regions in the network in which the node’s connection was high were considered clusters. The gene expression network was constructed at β = 12 to ensure scale-free topology (R^2^ ≥ 0.80). If the scale-free topology fit index for the reference dataset reaches values >0.8 for low powers (<30), the topology of the network is considered scale-free, and therefore, there are no batch effects [40,41,42,43]. Moreover, metabolic and signaling pathways enrichment in the ceRNA regulatory network and its subnets were identified using STRING, DAVID, and PANTHER websites.

## 3. Results

### 3.1. Identified DE miRNAs and mRNAs from Comparative Transcriptome Analysis

To construct the circRNA–lncRNA–miRNA–mRNA ceRNA regulatory network and identify molecular mechanisms of IMF and fat metabolism in beef cattle, comparative transcriptome profiles of LDM tissue were investigated from five beef cattle breeds during 10 pairwise comparisons. For this, RNA-Seq datasets were selected from the GEO database and used as experimental data. Due to differential expression analyses, 1792, 11,675, 10,864, 11,254, 11,111, 10,989, 11,353, 1394, 1231, and 1698 DE RNAs were obtained with the threshold of an FC ≥ 1 and ≤−1, and an FDR < 0.05 for Angus vs. Chinese Simmental, Angus vs. Luxi, Angus vs. Nanyang, Angus vs. Shandong Black, Chinese Simmental vs. Luxi, Chinese Simmental vs. Nanyang, Chinese Simmental vs. Shandong Black, Luxi vs. Nanyang, Luxi vs. Shandong Black, and Nanyang vs. Shandong Black comparison groups, respectively (Appendix A). Finally, 386 DE mRNAs and 11 miRNAs were identified in the Venn analysis as common DE RNAs between eight pairwise comparative groups and Gene Set 1 (Appendix A). Differentially expressed miRNAs are shown in Table 2.

### 3.2. Literature Mining and Determining of Main Gene List

Based on literature mining, 34 candidate circRNAs, 57 candidate lncRNAs, and 4 candidate miRNAs were identified (Appendix A) and considered for subsequent analyses. Finally, a list of the main RNAs list was obtained by integrating Appendix A from comparative transcriptome analyses and literature mining.

### 3.3. Gene Ontology and KEGG Pathway Enrichment Analysis of DE mRNAs

The GO analysis was performed based on the biological process (BP), molecular function (MF), and cellular component (CC) to explore relevant biological functions of DE mRNAs related to IMF and fat metabolism in beef cattle. Forty-eight biological processes were enriched based on DE mRNAs, of which the 10 top BP were metabolic process, cellular protein metabolic process, cellular nitrogen compound biosynthetic process, muscle cell differentiation, peptide metabolic process, carboxylic acid metabolic process, actin filament-based process, actin cytoskeleton organization, muscle structure development, and oxidation-reduction process (Figure 2A). Furthermore, DE mRNAs were involved in 13 significant molecular functions (MF), including enzyme binding, cytoskeletal protein binding, structural molecule activity, actin binding, kinase binding, protein kinase binding, actin filament binding, actinin binding, alpha-actinin binding, chaperone binding, muscle alpha-actinin binding, structural constituent of muscle, and kinase inhibitor activity (Figure 2B). Regarding cellular components (CC), 28 terms were identified, of which mitochondrion, myofibril, sarcomere, actin cytoskeleton, mitochondrial membrane, I band, Z disc, mitochondrial inner membrane, polymeric cytoskeletal fiber, and sarcolemma were the 10 top terms related to IMF in beef cattle (Figure 2C). Moreover, concerning the KEGG pathway analysis of DE mRNAs related to IMF and fat metabolism, 25 pathways were identified (Figure 3). Based on functional enrichment analysis, metabolic pathways, carbon metabolism, oxidative phosphorylation, regulation of actin cytoskeleton, citrate cycle (TCA cycle), as well as calcium, cGMP-PKG, thyroid hormone, oxytocin, and HIF-1 signaling pathways, were the top 10 highly associated pathways in IMF.

### 3.4. PPI Network Construction and Hub Genes Determining

The protein–protein interactions and PPI network construction were accomplished using the STRING database, indicating interactions between mRNAs based on biological and biochemical functions. The PPI network obtained had 370 nodes (mRNAs) and 2774 edges (interactions) (Appendix A). Furthermore, the hub mRNAs/genes were considered based on a higher connectivity rate in the PPI network, which included *ACTR3B*, *ASH1L*, *ATP2A2*, *BAG3*, *BIRC6*, *BRWD1*, *CKM*, *CTNND1*, *DNAJB4*, *EHBP1L1*, *HSPB7*, *LMOD2*, *MYOZ1*, *PDLIM1*, *RYR1*, *UTRN*, *VPS13C*, *VPS13D*, *VWF*, and *WDFY3*.

### 3.5. Reconstruction of circRNA–lncRNA–miRNA–mRNA ceRNA Regulatory Network

To discover how circRNAs and lncRNAs affect gene expression by targeting regulatory miRNAs and mRNAs, a circRNA–lncRNA–miRNA–mRNA ceRNA regulatory network was constructed with an integrated interaction between regulatory RNAs. Based on knowledge of extracted interaction data from the STRING database and other related original papers (literature mining), a circRNA–lncRNA–miRNA–mRNA ceRNA regulatory network that involved 455 nodes and 3056 edges was reconstructed. These 455 nodes included 34 circRNAs, 57 lncRNAs, 15 DE miRNAs, and 349 DE mRNAs, which were included in the regulatory network (Figure 4). In addition to hub mRNAs/genes, based on their connections in the ceRNA regulatory network, bta-miR-1296, bta-miR-365-1, and bta-miR-378d were defined as hub miRNAs. Additionally, MSTRG.10337 lncRNA was defined as a hub lncRNA due to its interaction in the ceRNA network.

### 3.6. Clustering Analysis of the circRNA–lncRNA–miRNA–mRNA ceRNA Regulatory Network

After the construction of the circRNA–lncRNA–miRNA–mRNA ceRNA regulatory network, clustering analysis was performed using the MCODE plugin of Cytoscape [39]. Output consisted of 11 subnets, some of which were sub-clusters of other larger clusters. After these subnets were removed, the total number of subnets decreased to 7, and included 16 circRNA, 43 lncRNA, 7 miRNA, and 237 DE mRNA (considering repeated nodes). In addition, subnets 1 to 7 had 147, 135, 123, 112, 50, 34, and 32 nodes, respectively (Figure 5, Figure 6, Figure 7, Figure 8, Figure 9, Figure 10 and Figure 11).

## 4. Discussion

Intramuscular fat (IMF) content is a highly complex trait associated with meat color, tenderness, juiciness, and flavor [7]. As adipogenesis affects the size and number of adipocytes in beef cattle, it can profoundly affect IMF. Moreover, muscle fiber diameter has a significant positive correlation with carcass traits [44]. For example, thicker muscle fiber diameter reduces meat tenderness [7]. Moreover, the type of muscle fiber and its fatty acid structure is highly related to IMF and various metabolic characteristics of meat in beef cattle [45]. Unsaturated fatty acids (UFAs), especially polyunsaturated fatty acids (PUFAs), affect meat taste and flavor [7]. With the continuous development and widespread application of high-throughput sequencing technologies, such as RNA-Seq, many studies have been performed on protein-coding RNAs (mRNAs) and miRNAs, without considering other molecular species of RNAs such as circRNAs and lncRNAs. Although mRNAs constitute only ~2% of the mammalian genome, considering interactions among regulatory agents such as lncRNAs can provide important insights into mechanisms involved in IMF and fat metabolism in beef cattle [46]. After the competitive endogenous RNA (ceRNA) hypothesis was proposed by Salmena et al. [11], many studies were conducted to explain how ceRNA (lncRNAs and, recently, circRNAs) function. However, the regulatory network that connects non-coding and coding RNAs has not been well explained. Perhaps lncRNAs and circRNAs compete with endogenous RNA by sponging to a certain miRNA to regulate the miRNA’s target mRNA/gene. Attempts to construct ceRNA networks have been made in animal science [12,15,47]. In this study, we identified 386 DE mRNAs, 15 DE miRNAs, 34 circRNAs, and 57 lncRNAs in the integrated analysis (RNA-Seq analysis and literature mining) of datasets by comparing five cattle breeds in 10 pairwise comparisons. Then, we recognized potential circRNA–lncRNA–miRNA–mRNA ceRNA interactions involved in IMF and fat metabolism in beef cattle. Finally, we constructed a circRNA–lncRNA–miRNA–mRNA ceRNA regulatory network with 455 nodes and 3056 edges. We also detected seven candidate subnets involved in IMF that included 237 mRNAs, 7 miRNAs, 16 circRNAs, and 43 lncRNAs, (with no repeated nodes), respectively, as well as subnets 1 to 7 that included 147, 135, 123 112, 50, 34, and 32 nodes, and 496, 744, 269, 152, 71, 48, and 33 edges, respectively. Information about differentially expressed RNAs and their interactions in subnets of the ceRNA regulatory network between eight pairwise comparative groups associated with IMF in beef cattle breeds was presented in Appendix A. Hub-hub mRNAs/genes were detected based on a higher degree of connectivity and repetition between the main ceRNA regulatory network and subnets. Furthermore, our results classified the GO functional annotation terms of DE RNAs into three groups: biological process (48 terms), cellular component (28 terms), and molecular function (13 terms), with most identified metabolic and signaling pathways associated with the structure of muscle cells, fat metabolism, and intracellular energy pathways.

In this study, we identified the *MCU*, *CYB5R1*, and *BAG3* genes that were mostly down-regulated between the 10 comparative groups and were involved with the metabolic process, actin filament-based process, actin cytoskeleton organization, oxidation-reduction process, actin filament organization, chaperone binding, myofibril, mitochondrion, sarcomere, actin cytoskeleton, mitochondrial membrane, I band, Z disc, mitochondrial inner membrane, mitochondrial protein complex, inner mitochondrial membrane protein complex, and actomyosin. The *MCU* (Mitochondrial inner membrane calcium uniporter) gene is a protein-encoding that interacts with mitochondrial calcium uptake and is involved in calcium homeostasis in mitochondria. Mitochondrial calcium homeostasis has a critical role in cellular physiology, including the regulation of cell bioenergetics, cytoplasmic calcium signals, and activation of cell death pathways. This gene is also involved in facilitating the calcium flow in cardiomyocytes during systole, regulating glucose-dependent insulin secretion, muscle size, metabolic pathways of prion disease, and the calcium signaling pathway [48,49]. The *CYB5R1* (Cytochrome B5 Reductase 1) gene is associated with meat tenderness and oleic acid percentage in Jiaxian Red and Japanese Black cattle breeds, serving as an electron source for stearoyl-CoA desaturase during fatty acid desaturation [50,51]. *BAG3* (BCL2-associated athanogene 3) gene was identified as a hub-hub gene mostly expressed in heart and skeletal muscle tissue. The protein encoded by this gene is involved in facilitating autophagy, inhibition of apoptosis by binding to B-cell lymphoma 2, providing structural support for the sarcomere by attaching actin to the Z disk, and linking the α-adrenergic receptor with the L-type Ca^2+^ channel [52,53].

In subnet 1, the *UTRN* gene (Utrophin) is an autosomal protein-coding gene, and the protein encoded by this gene is smaller than dystrophin and localized to the sarcolemmal post-synaptic membrane at the neuromuscular junction and myotendinous junction in mature muscle fibers. Utrophin is highly similar to dystrophin and can complete dystrophin’s function. For example, utrophin interacts with the protein complex associated with dystrophin to complete a link from the cytoskeleton through the membrane and to the extracellular matrix. Utrophin expression is typically high in developing muscle, but in mature muscle is enriched at the neuromuscular junction. Expression of utrophin decreases as muscle fiber matures, whereas dystrophin is expressed in mature muscle fibers [54]. Significant GO terms associated with the IMF affected by the *UTRN* gene included muscle structure development, muscle system process, actin binding, cytoskeletal protein binding, enzyme binding, and the actin cytoskeleton. Regarding the activation role of bta-miR-1-2 for the *UTRN* gene, it is understandable to be down-regulated. In this study, *LMOD2* and *RYR1* genes were jointly involved in nine GO terms, including muscle structure development, muscle cell differentiation, muscle cell development, striated muscle cell differentiation, striated muscle cell development, muscle system process, muscle contraction in biological processes, myofibril and sarcomere in cellular components. Moreover, gene *LMOD2* was enriched in terms of actin filament-based process, actin cytoskeleton organization, actomyosin structure organization, myofibril assembly, actin filament organization, sarcomere organization, actin polymerization or depolymerization, actin binding, cytoskeletal protein binding, actin cytoskeleton, polymeric cytoskeletal fiber, A band, actin filament, and M band. The *RYR1* gene was involved in striated muscle tissue development, muscle organ development, skeletal muscle tissue development, muscle fiber development, I band, Z disc, sarcolemma, sarcoplasm, sarcoplasmic reticulum, and sarcoplasmic reticulum membrane (Figure 2).

Leiomodin (LMOD) is an actin-binding protein together with a homolog protein of tropomodulin (TMOD) present at the slow-growing (pointed) end of the actin filament that regulates filament lengths [55]. *LMOD1* gene is expressed in smooth muscle and extraocular muscle tissues, whereas *LMOD2* and *LMOD3* are mainly in cardiac and skeletal muscle tissues. *LMOD2* expressed in cardiac muscle tissue is higher than in skeletal muscle and may have a major role in the maintenance of thin filaments of cardiac muscle tissue [56]. The protein encoded by *CKM* (Creatine kinase, M-type) gene is the most abundant transcript in skeletal muscle tissue [57]. Creatine kinase (CK) is a major enzyme in energy metabolism that catalyzes the transportation of phosphate between ATP molecules and various phosphagens. Isoenzymes of CK are present in various tissues, including skeletal and heart muscle, spleen, thyroid, and blood, with important roles in energy transfer in many tissues. Expression of the *CKM* gene differs among anatomical muscle groups. For example, the level of CKM protein as well as the enzyme byproduct, creatine phosphate, is two or three times higher in fast-versus slow-twitch muscle [58]. The *RYR1* (Ryanodine Receptor 1) gene encoded a calcium ion (Ca^2+^) release channel located in the sarcoplasmic reticulum membrane in the skeletal muscle and activated by CaV1.1 voltage sensor proteins during excitation-contraction (EC) [59], is a 2.2 megadalton molecule responsible for calcium gating in the sarcoplasmic reticulum. Ryanodine receptor 1 (*RYR1*) gene-related pathogenic variations are the most frequent causes of congenital myopathies [60]. RYR1-related myopathies (RYR1-RM) are variable in intensity and include a wide disease spectrum of appointed and emerging phenotypes related to dominant and recessive inheritance patterns [61]. A selective pattern and gradient of intramuscular fatty infiltration (IMFI) can intensify RYR1-RM. High IMFI and loss of muscle mass can cause skeletal muscle dysfunctions and increase disease severity, such as muscular dystrophies and sarcopenic adults [62].

In subnet 2, bta-miR-133a-1 and bta-miR-133a-2 miRNAs suppressed the MSN gene and were down-regulated for LDM tissue between comparative groups. Moesin, the protein encoded by the *MSN* gene, is a member of the Ezrin-Radixin-Moesin (ERM) family of proteins. ERMs are adaptor molecules that are essential for organizing specified membrane domains and implicated in diverse basic biological processes, including regulation of cell shape, motility, and signaling. ERMs also regulate the structural stability of the cell cortex by connecting the actin cytoskeleton to plasma membrane proteins via an N-terminal FERM domain and a C-terminal actin-binding domain [63]. *ACTR3B* was a hub-hub gene and down-regulated in LDM tissue. *ACTR3B* gene encodes a member of the actin-related proteins (ARP), which form multiprotein complexes such as actin-related protein (ARP) 2/3 complex and share 35–55% amino acid identity with conventional actin. As the protein coded by this gene is present in the APR2/3 complex, it may have a role in the organization of the actin cytoskeleton and function as an ATP-binding component of the Arp2/3 complex, which is involved in the regulation of actin polymerization. This complex is a regulatory protein in the actin cytoskeleton and induces cell-shape change and motility [64,65]. Actin filament-based process, actin cytoskeleton organization, actin binding, cytoskeletal protein binding and actin cytoskeleton were the highly associated GO terms affected by the *ACTR3B* gene.

In subnet 3, *ATP2A2*, *MYOZ1*, and *EHBP1L1* were hub-hub genes. Myozenin 1, known as calsarcin-2 protein (CS2), is encoded by the *MYOZ1* gene and mainly expressed in fast-twitch muscles, a 299 amino acid nuclear protein that interacts with the Z-disc protein, α-actinin, filamin 2, and PP2B (calcineurin), and effectively forms bridges between proteins and muscle fibers, participating in muscle sarcomere microstructure [66]. According to interactions of the protein encoded by this gene, *MYOZ1* may be involved in Z-line assembly and myofibril formation in striated skeletal muscles. According to reports from immunofluorescence experiments, calsarcin (CS) family proteins are specifically located on the Z line [67]. Because the *MYOZ1* gene is closely related to muscle formation, mutations in the *MYOZ1* gene can be associated with muscular dystrophy and neuromuscular myopathy [68]. The *MYOZ1* gene is involved in 22 GO terms, including the actin filament-based process, actin cytoskeleton organization, muscle structure development, muscle cell differentiation, muscle cell development, striated muscle cell differentiation, actomyosin structure organization, muscle system process, myofibril assembly, striated muscle tissue development, muscle organ development, sarcomere organization, skeletal muscle tissue development, and regulation of calcium-mediated signaling in the biological process, actin binding, and cytoskeletal protein binding in the molecular function, and myofibril, sarcomere, actin cytoskeleton, I band, and Z disc in the cellular component.

The *ATP2A2* gene is mainly expressed in cardiac and type II skeletal muscle and encodes SERCA2, one of the SERCA Ca(^2+^)-ATPases, which is an intracellular calcium pump in the sarcoplasmic or endoplasmic reticulum in skeletal muscle cells [69]. SERCAs regulate cytosolic calcium homeostasis and coordinate gene expression and muscle cell function. SERCA2 specifically catalyzes the hydrolysis of ATP along with the transport of calcium from the cytosolic region to the lumen of the sarcoplasmic reticulum and is involved in the regulation of the contraction-relaxation cycle [70]. Interestingly, this gene was in the majority of enriched pathways, including the calcium signaling pathway, cGMP-PKG signaling pathway, adrenergic signaling in cardiomyocytes, thyroid hormone signaling pathway, cardiac muscle contraction, and hypertrophic cardiomyopathy, suggesting a possible major regulatory role. Other noteworthy hub-hub genes, including *CKM*, *RYR1*, and *VWF,* presented lower expression in LDM tissue and have critical roles in metabolic pathways, oxytocin signaling pathway, ECM-receptor interaction, prion disease, and calcium signaling pathway. In this study, the *RYR1* was involved in 19 GO terms, including muscle structure development, muscle cell differentiation, muscle cell development, striated muscle cell differentiation, striated muscle cell development, muscle system process, muscle contraction, striated muscle tissue development, muscle organ development, skeletal muscle tissue development, and muscle fiber development, myofibril, sarcomere, I band, Z disc, sarcolemma, sarcoplasm, sarcoplasmic reticulum, and sarcoplasmic reticulum membrane.

Signaling pathways related to this subnet included the cGMP-PKG, thyroid hormone, and calcium signaling pathways. Cyclic GMP (cGMP) is the intracellular second messenger that is produced by guanylate cyclase (GC) and regulates a broad range of biological processes. Intracellular cGMP exerts its physiological action through three forms of cGMP-dependent protein kinase (PKG), cGMP-regulated phosphodiesterases (PDE2, PDE3), and cGMP-gated cation channels, among which PKGs may be the primary mediator. cGMP-PKG signaling pathway mediates many processes, such as regulation of relaxation and contraction of vascular smooth muscle cells, anti-cardiac hypertrophy, anti-atherosclerosis, and anti-vascular injury/restenosis [71]. The thyroid hormone signaling pathway has a wide range of functions in terms of individual development, maintenance of homeostasis, cell proliferation and differentiation, and glucose metabolism. Thyroid hormones thyroxine (T3) and triiodothyronine (T4) are produced by the thyroid gland and have a main role in maintaining growth and development. Although T4 is the main hormone in the blood, it is converted to the more active hormone T3 within cells. T3 binds to nuclear thyroid hormone receptors (TRs), which act as a ligand-dependent transcription factor and control the expression of target genes (genomic action). Non-genomic mechanisms of action begin at the integrin receptor. Plasma membrane alpha(v)beta(3)-integrin has distinct binding sites for T3 and T4. A single binding site binds T3 and activates the phosphatidyl 3-kinase (PI3K) pathway. The other binding site binds both T3 and T4 and activates the ERK1/2 MAP kinase pathway [72,73]. Ca^2+^ is also a very versatile intracellular signaling molecule capable of regulating many processes. Its distribution in intracellular and extracellular spaces necessitates specialized pumps and channels for its function and mobility, as well as the effect of cell depolarization or repolarization. In addition, the amount and duration of Ca^2+^ influx determine the type and duration of its effect on intracellular signaling. Ca^2+^ signaling not only controls intracellular regulation but also appears to contribute to remote or even organismal signal propagation and physiological response regulation. The calcium signaling pathway is shaped by an intimate interaction of channels and transporters, and important individual components have been identified and characterized. This is translated into defined downstream responses by an elaborate toolbox of Ca^2+^-binding proteins, many of which act as Ca^2+^ sensors [74].

In subnet 4, *GLUL* is a protein-coding gene that encodes glutamate-ammonia ligase and belongs to the glutamine synthetase family. This protein catalyzes glutamine synthesis from glutamate and ammonia in an ATP-dependent reaction. This protein has a role in ammonia and glutamate detoxification, acid-base homeostasis, cell signaling, cell proliferation, and also the biosynthesis of several amino acids, pyrimidines, and purines by catalyzing the glutamine synthesis reaction [75,76]. The GO terms enriched for this gene included metabolic process, cellular protein metabolic process, carboxylic acid metabolic process, and mitochondrion. *ASH1L* gene encoded a histone methyltransferase that belongs to the Trithorax group of transcriptional activators. This gene apparently has a role in the positive regulation of gene expression and is positively correlated with myoblast fusion and myogenesis [77]. This gene is involved in both the metabolic process and the cellular protein metabolic process. Moreover, metabolic pathways enriched for *GLUL* and *ASH1L* genes included the metabolic pathways, biosynthesis of amino acids and glyoxylate, and dicarboxylate metabolism.

In subnets 5 and 6, *VPS13C* and *VPS13D* hub-hub genes are members of the vacuolar sorting protein-13 family of proteins that have four isomers: *VPS13C*, *VPS13B*, *VPS13A*, and *VPS13D*. The VPS13 family is involved in the transport of membrane proteins between the trans-Golgi network and the pre-vacuolar compartment, as well as lipid transport and mitophagy [78]. Mitophagy is crucial in regulating cell health, mitochondrial size, and homeostasis [79]. *VPS13C* and *VPS13D* genes are involved in terms of regulation of mitophagy, positive regulation of mitophagy, mitochondrion, mitochondrial membrane, and mitochondrial outer membrane. The *BRWD1* gene encodes a member of the WD repeat protein family, which contains two bromodomains and multiple WD repeats. WD repeats are minimally conserved regions of almost 40 amino acids typically bracketed by gly-his and trp-asp (GH-WD) residues that may facilitate the formation of heterotrimeric or multiprotein complexes. Members of this family are involved in various cellular processes such as cell cycle progression, signal transduction, apoptosis, regulation of chromatin remodeling, and gene expression [80].

In subnet 7, we detected bta-miR-378d that suppressed the *RTN2* hub gene. The *RTN2* (Reticulon 2) gene is defined as a hub gene that inhibits amyloid precursor protein processing, probably by blocking BACE1 activity. This gene enhances the trafficking of the glutamate transporter SLC1A1/EAAC1 from the endoplasmic reticulum to the cell surface and has a role in the translocation of SLC2A4/GLUT4 from intracellular membranes to the cell membrane, facilitating glucose uptake into the cell [81].

We used a computational approach with the construction of a circRNA–lncRNA–miRNA–mRNA ceRNA regulatory network using identified expression profiles of regulatory RNAs. Spatiotemporal differential expression in LDM tissue supports the potential role of RNAs; furthermore, it can have a major role in identifying candidate regulatory RNAs in transcriptional regulation involved in IMF and fat metabolism. However, further efforts are necessary to identify specific biological functional roles of RNA regulatory subnets associated with meat quality traits. A typical explanation for inconsistencies and limitations in our results compared to other studies was differences in the molecular techniques, differences in LDM tissue, time and environmental conditions of sampling from different beef cattle breeds, and bioinformatics tools. Nevertheless, the integration of various regulatory RNAs based on ceRNA regulatory networks, plus circRNA, lncRNA, miRNA, and mRNA interactions, provided novel insights into molecular biological processes.

## 5. Conclusions

The present study used a method that is novel in animal science to combine various regulatory RNAs as an integrated network related to IMF and fat metabolism in beef cattle. Integrating transcriptome profiles from differential expression analysis of five beef cattle breeds (i.e., Angus, Chinese Simmental, Luxi, Nanyang, and Shandong Black) to provide hub RNAs with differential expression levels resulted in the identification of 34 circRNAs, 57 lncRNAs, 15 miRNAs, and 374 mRNA/genes involved in IMF and lipid metabolism. According to the circRNA–lncRNA–miRNA–mRNA ceRNA regulatory network, 7 significant subnets with a total of 16 circRNAs, 43 lncRNAs, 7 miRNAs, and 237 mRNAs/genes were identified as being involved in major biological molecular mechanisms, including metabolic pathways, muscle structure development, oxidation-reduction process, protein kinase binding, actin filament binding, mitochondrial inner membrane, calcium, cGMP-PKG, thyroid hormone, and oxytocin signaling pathways. In conclusion, performing comparative transcriptome and ceRNA regulatory network analyses between various breeds for LDM tissue generated novel insights for breeding strategies on the meat quality attributes of beef cattle.

## Figures and Tables

**Figure 1 animals-13-02598-f001:**
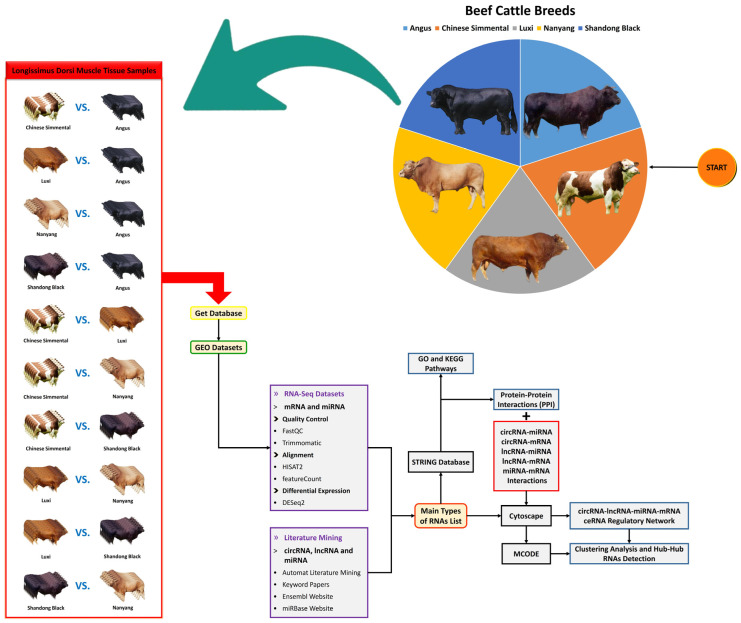
Schematic of the workflow used to construct the circRNA–lncRNA–miRNA–mRNA ceRNA regulatory network of intramuscular fat (IMF) content in beef cattle. Regulatory RNAs were obtained from RNA-Seq data analyses and literature mining. The protein–protein interaction network (PPI), gene regulatory network (GRN), and ceRNA regulatory network were prepared using STRING and Cytoscape software.

**Figure 2 animals-13-02598-f002:**
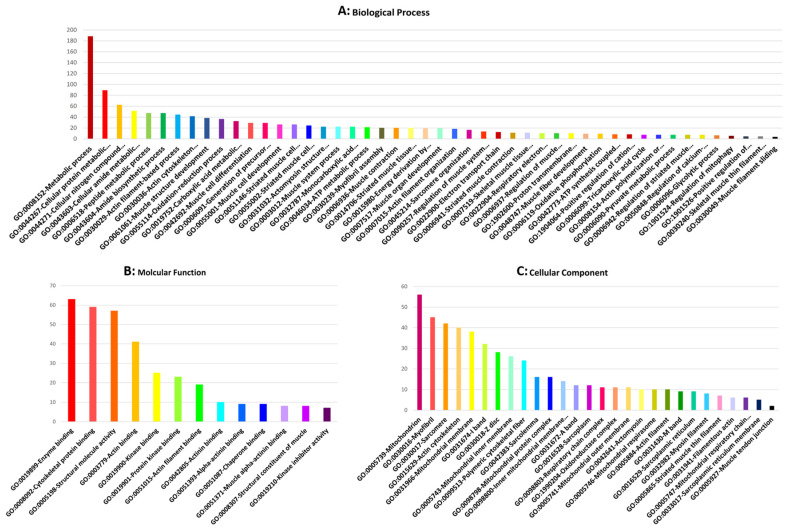
Top significant mRNA/gene ontology (GO) terms enriched using mRNAs/genes associated with intramuscular fat content in beef cattle.

**Figure 3 animals-13-02598-f003:**
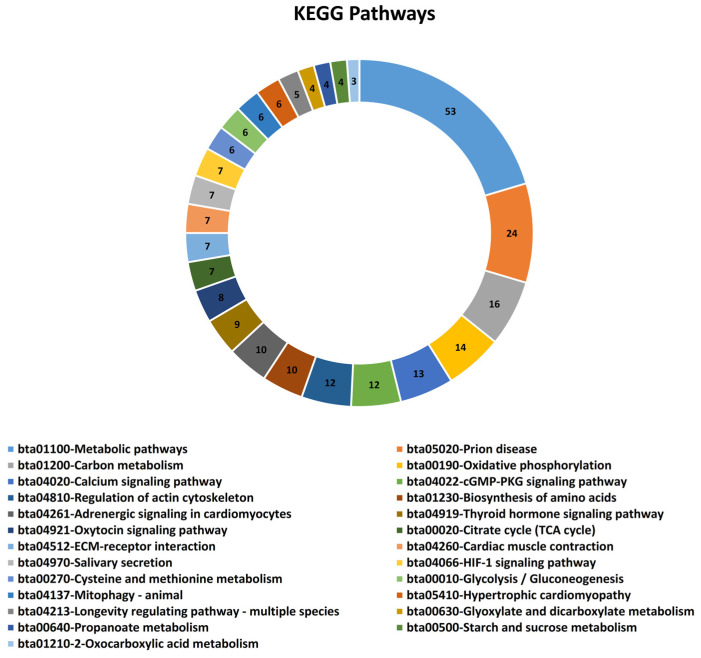
Top KEGG pathways enriched using significant mRNAs/genes associated with intramuscular fat content in beef cattle.

**Figure 4 animals-13-02598-f004:**
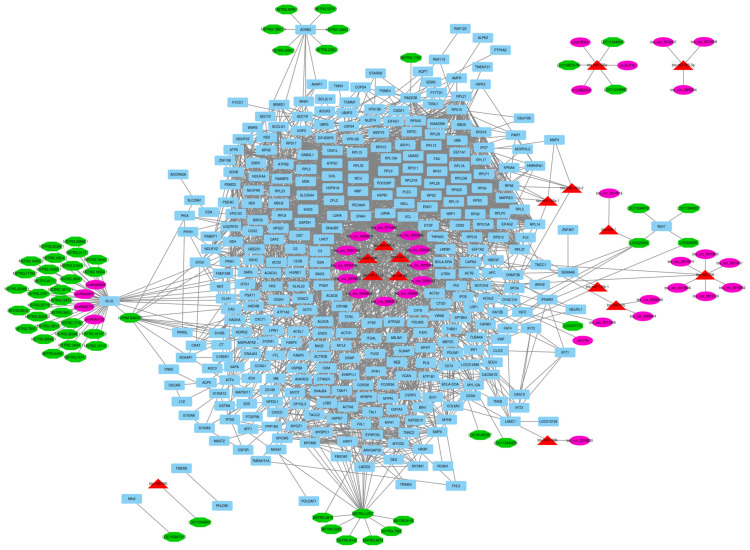
ceRNA regulatory network: 34 circRNAs, 57 lncRNAs, 15 DE miRNAs, and 349 DE mRNAs were identified in an interacted network. Circular nodes represent circRNAs, octagonal nodes represent lncRNAs, triangle nodes represent miRNAs, and quadrilateral nodes represent mRNAs/genes. Black edges indicate interactions between nodes.

**Figure 5 animals-13-02598-f005:**
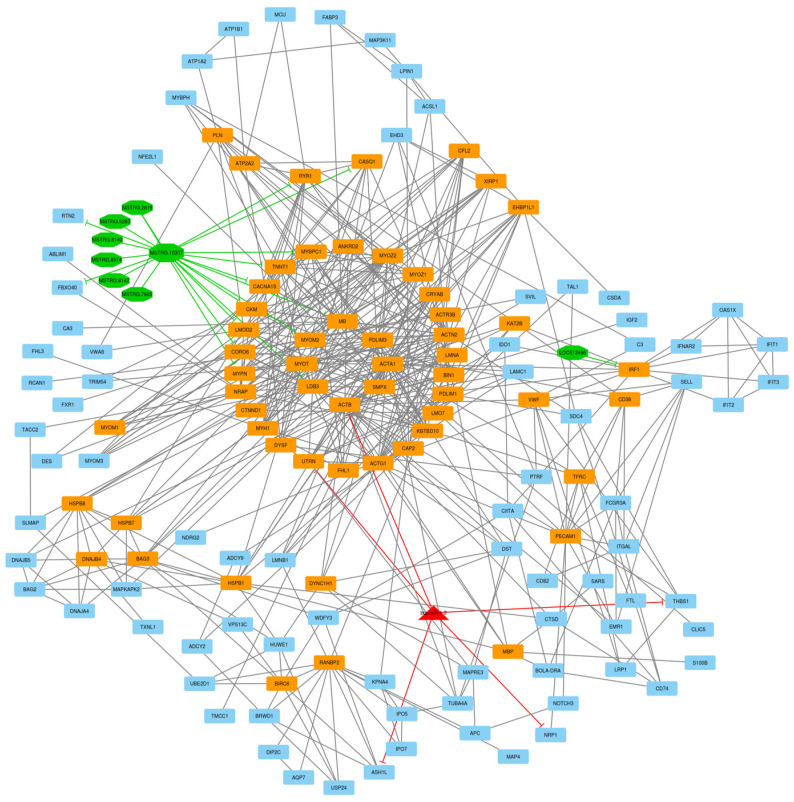
Subnet 1: 8 lncRNAs, 1 DE miRNA, and 138 DE mRNAs in an interacted network were identified. In this subnet, octagonal nodes represent lncRNAs, triangle nodes represent miRNAs, and quadrilateral nodes represent mRNAs/genes. The big green octagonal nodes and orange quadrilateral represent hub lncRNAs and mRNAs/genes involved in the subnet, respectively. Edges indicate interactions; black edges represent mRNA–mRNA interactions, green edges represent lncRNA–mRNA and lncRNA–lncRNA interactions, and red edges represent miRNA–mRNA interactions.

**Figure 6 animals-13-02598-f006:**
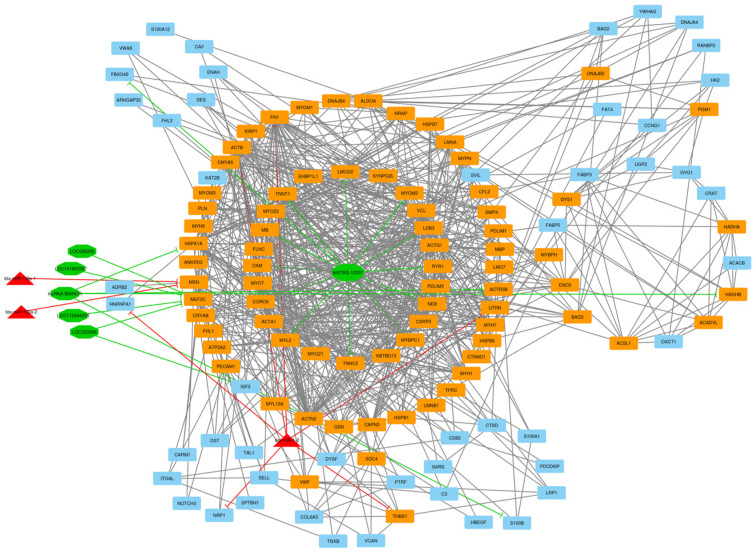
Subnet 2: 6 lncRNAs, 3 DE miRNA, and 126 DE mRNAs in an interacted network were identified. In this subnet, octagonal nodes represent lncRNAs, triangle nodes represent miRNAs, and quadrilateral nodes represent mRNAs/genes. The big green octagonal nodes and orange quadrilateral represent hub lncRNAs and mRNAs/genes involved in the subnet, respectively. Edges indicate interactions; black edges represent mRNA–mRNA interactions, green edges represent lncRNA–mRNA interactions and red edges represent miRNA–mRNA interactions.

**Figure 7 animals-13-02598-f007:**
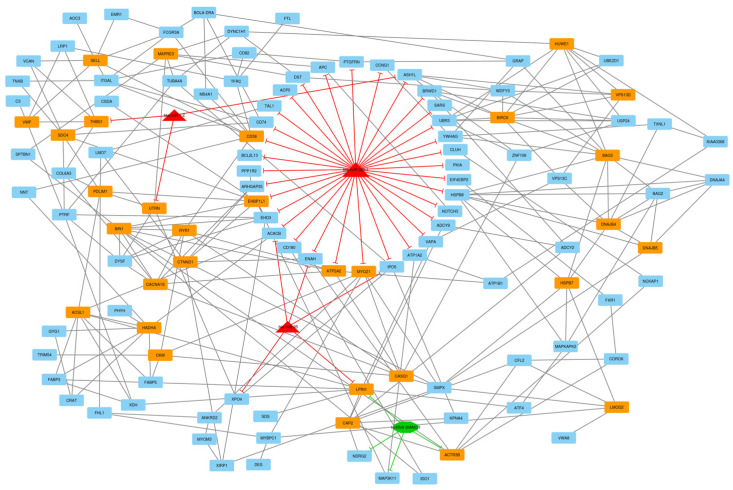
Subnet 3: 1 lncRNA, 3 DE miRNA, and 119 DE mRNAs in an interacted network were identified. In this subnet, octagonal nodes represent lncRNAs, triangle nodes represent miRNAs, and quadrilateral nodes represent mRNAs/genes. Big red triangle nodes and orange quadrilateral represent hub miRNAs and mRNAs/genes involved in the subnet, respectively. Edges indicate interactions; black edges represent mRNA–mRNA interactions, green edges represent lncRNA–mRNA interactions and red edges represent miRNA–mRNA interactions.

**Figure 8 animals-13-02598-f008:**
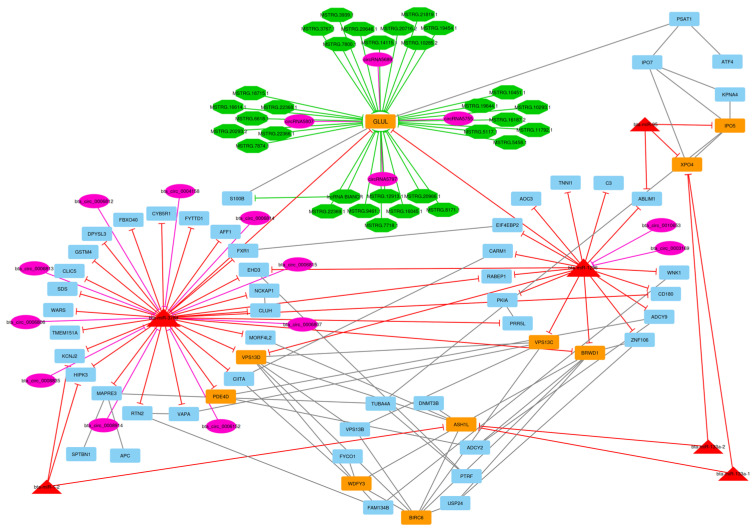
Subnet 4: 16 circRNAs, 31 lncRNA, 6 DE miRNA, and 59 DE mRNAs in an interacted network were identified. In this subnet, circular nodes represent circRNAs, octagonal nodes represent lncRNAs, triangle nodes represent miRNAs, and quadrilateral nodes represent mRNAs/genes. Big red triangle nodes and orange quadrilateral represent hub miRNAs and mRNAs/genes involved in the subnet, respectively. Edges indicate interactions; black edges represent mRNA–mRNA interactions, purple edges represent circRNA–mRNA interactions, green edges represent lncRNA–mRNA interactions and red edges represent miRNA–mRNA interactions.

**Figure 9 animals-13-02598-f009:**
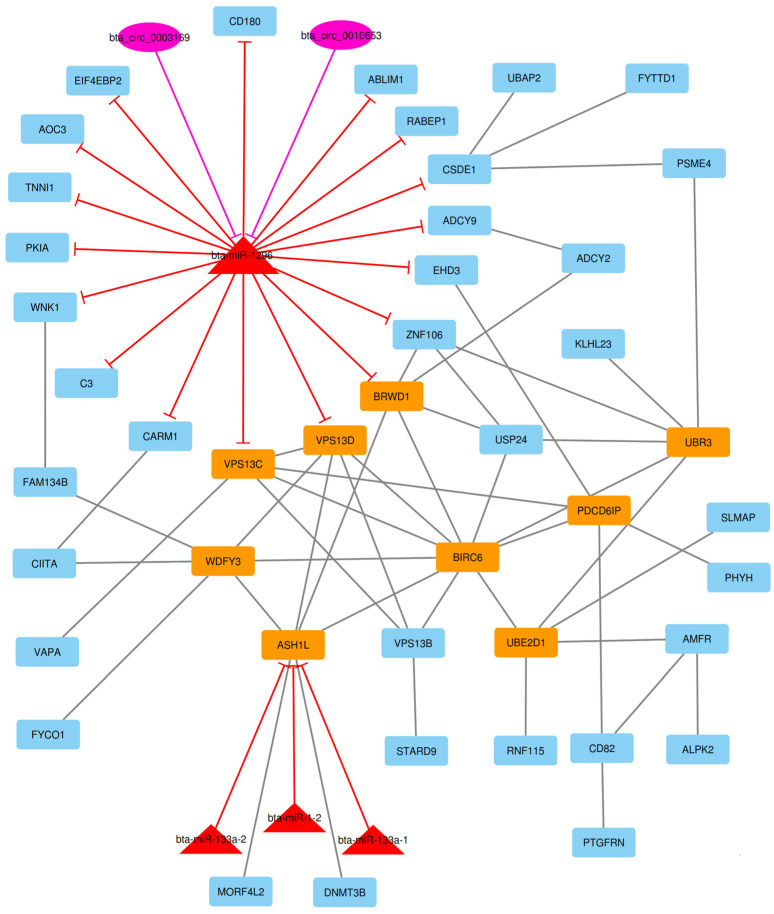
Subnet 5: 2 circRNAs, 4 DE miRNAs, and 44 DE mRNAs in an interacted network were identified. In this subnet, circular nodes represent circRNAs, triangle nodes represent miRNAs, and quadrilateral nodes represent mRNAs/genes. Big red triangle nodes and orange quadrilateral represent hub miRNAs and mRNAs/genes involved in the subnet, respectively. Edges indicate interactions; black edges represent mRNA–mRNA interactions, purple edges represent circRNA–mRNA interactions and red edges represent miRNA–mRNA interactions.

**Figure 10 animals-13-02598-f010:**
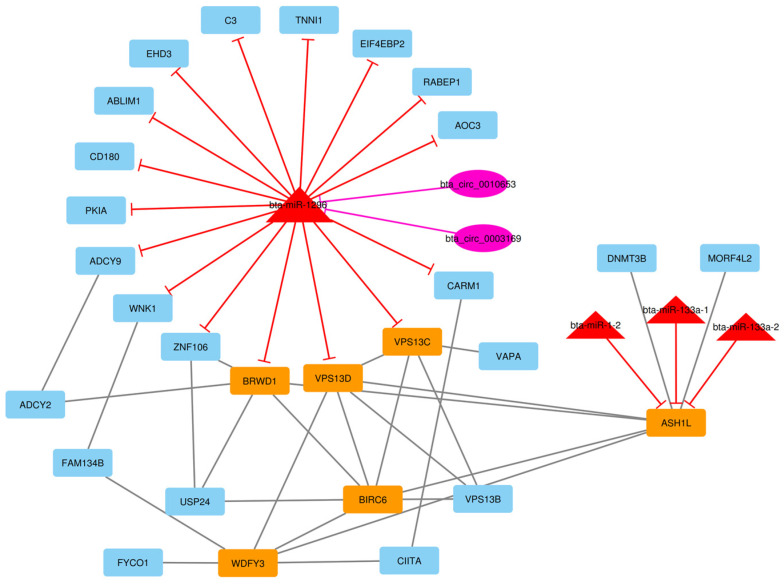
Subnet 6: 2 circRNAs, 4 DE miRNAs, and 28 DE mRNAs in an interacted network were identified. In this subnet, circular nodes represent circRNAs, triangle nodes represent miRNAs, and quadrilateral nodes represent mRNAs/genes. Big red triangle nodes and orange quadrilateral represent hub miRNAs and mRNAs/genes involved in the subnet, respectively. Edges indicate interactions; black edges represent mRNA–mRNA interactions, purple edges represent circRNA–mRNA interactions and red edges represent miRNA–mRNA interactions.

**Figure 11 animals-13-02598-f011:**
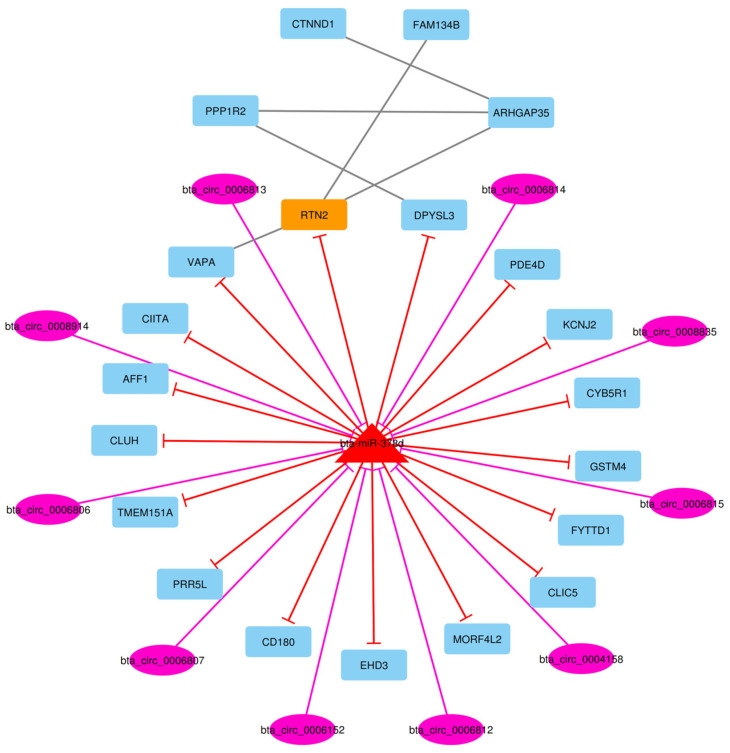
Subnet 7: 10 circRNAs, 1 DE miRNA, and 21 DE mRNAs in an interacted network were identified. In this subnet, circular nodes represent circRNAs, triangle nodes represent miRNAs, and quadrilateral nodes represent mRNAs/genes. Big red triangle nodes and orange quadrilateral represent hub miRNAs and mRNAs/genes involved in the subnet, respectively. Edges indicate interactions; black edges represent mRNA–mRNA interactions, purple edges represent circRNA–miRNA interactions and red edges represent miRNA–mRNA interactions.

**Table 1 animals-13-02598-t001:** Summary of GEO accession numbers for RNA-Seq data sets of intramuscular fat (IMF) content of the five beef cattle breeds used in this study.

No.	GEO Accession	Platform	Breed	Sample	Citation
1	GSE171876	GPL26012 (Illumina NovaSeq 6000 (*Bos taurus*))	Angus	GSM5236003, GSM5236004, GSM5236005	Zheng et al. [23]
			Chinese Simmental	GSM5236006, GSM5236007, GSM5236008	
2	GSE161272	GPL15749 (Illumina HiSeq 2000 (*Bos taurus*))	Shandong Black	GSM4904154, GSM4904155, GSM4904156	Liu et al. [7]; Liu et al. [4]
			Luxi	GSM4904157, GSM4904158, GSM4904159	
3	GSE139102	GPL21659 (Illumina HiSeq 3000 (*Bos taurus*))	Nanyang	GSM4131022, GSM4131023, GSM4131024	Huang et al. [2]; Zhu et al. [6]

**Table 2 animals-13-02598-t002:** Differentially expressed miRNAs between eight pairwise comparative groups associated with IMF in beef cattle breeds.

miRNA Name	miRNA Locus			Fold Changes (FC)	*p*-Value	FDR
	BTA	miRNA Start	miRNA End			
bta-miR-95	6	114336289	114336369	−11.272055	7.39 × 10^−20^	3.06 × 10^−18^
bta-miR-1-2	24	34453354	34453464	−15.534093	3.98 × 10^−20^	1.76 × 10^−18^
bta-miR-125b-1	15	32763901	32763988	−9.7970735	1.38 × 10^−13^	1.91 × 10^−12^
bta-miR-1296	28	19489827	19489932	−7.0239642	0.0001303	0.0002563
bta-miR-133a-1	13	54735656	54735750	−19.140042	8.25 × 10^−59^	2.49 × 10^−56^
bta-miR-133a-2	24	34456691	34456777	−15.495943	1.16 × 10^−38^	1.86 × 10^−36^
bta-miR-193B	25	13248826	13248935	−11.619275	1.69 × 10^−21^	9.08 × 10^−20^
bta-miR-206	23	24567204	24567289	−17.307776	5.15 × 10^−45^	1.02 × 10^−42^
bta-miR-27b	8	81617234	81617337	−12.70512	1.79 × 10^−63^	6.65 × 10^−61^
bta-miR-365-1	25	13253584	13253670	−6.2762783	0.0021245	0.0035438
bta-miR-378d	19	8484500	8484590	−7.4127107	3.99 × 10^−7^	1.21 × 10^−6^

## Data Availability

The datasets used in this study are publicly available and can be accessed from the National Center for Biotechnology Information (NCBI) with BioProject numbers PRJNA721312, PRJNA676109, and PRJNA578388. Further details on accessing the data can be found on the NCBI website at https://www.ncbi.nlm.nih.gov/bioproject/ (accessed on 5 February 2023).

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
