# Peer review of "Integrated Comparative Transcriptome and circRNA-lncRNA-miRNA-mRNA ceRNA Regulatory Network Analyses Identify Molecular Mechanisms Associated with Intramuscular Fat Content in Beef Cattle"

_animals, 2023, doi:10.3390/ani13162598_

Round 1

Reviewer 1 Report

The title is descriptive of the content. It just feels a bit much for one manuscript. This could maybe rather go through as an review? Biochemically very interesting.

Summary is understandable and the results of the study could be useful in breeding selection in terms of intramuscular fat characteristics.

Abstract: Well written. This study describes a new approach in predicting the potential meat quality characteristics of different cattle breeds.
Some knowledge of genomic terms is needed to understand this manuscript.
Correction needed in line 39.

Introduction

The role of different types of RNA molecules were described in regulating the expression of genes involved in adiposal  related mechanisms influencing intra muscular fat characteristics. The aim of the project is clear.
LDM = longissimus dorsi muscle. Not a usual abbreviation. Not defined before.

Methods

The project design seems right. There is a duplication in the presentation of the test groups in the workflow figure 1. I am not used to the technique of literature mining, but the referral to different original investigators seems above board. I understand that databases are available for use. My knowledge on what software programs are the best to use, are very limited. I believe there are different ones for different purposes. I will not be able to judge.

Results

The massive amount of data is well presented and a very broad metabolic base were covered. It feels over whelming though. This is proof that biochemically the prediction of meat characteristics are very difficult and the regulation of the different mechanisms are very complex. Practically it seems nonviable to use these information for breeding selection. Environmental factors would influence expression so much that the genomic knowledge will only be useful on the macro scale. Non the less this research are very interesting and good fundamentally.

Discussion

Line 402 - needs reference

Lines 406 -411 - too long sentence

The discussion about the gene expression and down and up regulation of physiological properties in the cell is very long and probable good for an science handbook. The practical implementation of this knowledge for animal breeding will be more difficult. Some of the work is part of a literature study and for a scientific paper it should not be added. Maybe it was part of a graduate study and for publication one should be more selective.

Conclusions

Only last sentence was part of conclusion. Interesting concept.

Good

Author Response

Response to Reviewer 1 Comments

The authors greatly appreciate your insightful comments and have taken them into account to revise the manuscript. Changes are highlighted with yellow text.

 Comments and Suggestions for Authors:

  • The title is descriptive of the content. It just feels a bit much for one manuscript. This could maybe rather go through as an review? Biochemically very interesting.

Response: We are grateful for all your constructive comments. We have tried to define a comprehensive title that covers all parts of the article for the manuscript. In this regard, the title of most of the articles published in the field of animal science in the past few years are descriptively defined.

Summary

  • Summary is understandable and the results of the study could be useful in breeding selection in terms of intramuscular fat characteristics.

Response: We are grateful for your constructive comments.

Abstract

  • Well written. This study describes a new approach in predicting the potential meat quality characteristics of different cattle breeds.
  • Some knowledge of genomic terms is needed to understand this manuscript.
  • Correction is needed in line 39.

Response: We applied the mentioned changes in the revised manuscript.

Introduction

  • The role of different types of RNA molecules were described in regulating the expression of genes involved in adipose-related mechanisms influencing intra-muscular fat characteristics.
  • The aim of the project is clear.
  • LDM = longissimus dorsi muscle. Not a usual abbreviation. Not defined before.

Response: We defined longissimus dorsi muscle (LDM) changes in the Summary, Abstract, and Introduction sections in the revised manuscript.

Methods

  • The project design seems right. There is a duplication in the presentation of the test groups in the workflow figure 1. I am not used to the technique of literature mining, but the referral to different original investigators seems above board. I understand that databases are available for use. My knowledge on what software programs are the best to use, are very limited. I believe there are different ones for different purposes. I will not be able to judge.

Response:

  • We corrected the mentioned duplication in the presentation of the test groups in the workflow (Figure 1).
  • Literature mining led to collection of a significant and key list of genes that interact, together with genes identified in this study, highlighting metabolic-signaling pathways for intramuscular fat content in beef cattle. Literature mining was done to identify and collect all types of non-coding RNAs and also their interactions, enabling reconstruction of the ceRNA regulatory network, which can enhance understanding of gene regulation mechanisms related to intramuscular fat content in beef cattle. We have tried to use the most practical and newest software, as well as online websites.

> Ghafouri, F., Sadeghi, M., Bahrami, A., Naserkheil, M., Dehghanian Reyhan, V., Javanmard, A., Miraei-Ashtiani, R., Ghahremani, S., Barkema, H.W., Abdollahi-Arpanahi, R. and Kastelic, J.P. (2023). Construction of a circRNA-lincRNA-lncRNA-miRNA-mRNA ceRNA Regulatory Network Identifies Genes and Pathways Linked to Goat Fertility. Frontiers in Genetics, 14, 1195480.

> Reyhan, V.D., Sadeghi, M., Miraei-Ashtiani, S.R., Ghafouri, F., Kastelic, J.P. and Barkema, H.W. (2022). Integrated transcriptome and regulatory network analyses identify candidate genes and pathways modulating ewe fertility. Gene Reports, 28, 101659.

> Sadeghi, M., Bahrami, A., Hasankhani, A., Kioumarsi, H., Nouralizadeh, R., Abdulkareem, S. A., ... & Barkema, H. W. (2022). lncRNA–miRNA–mRNA ceRNA Network Involved in Sheep Prolificacy: An Integrated Approach. Genes, 13(8), 1295.

> Naserkheil, M., Ghafouri, F., Zakizadeh, S., Pirany, N., Manzari, Z., Ghorbani, S., ... & Min, K. S. (2022). Multi-omics integration and network analysis reveal potential hub genes and genetic mechanisms regulating bovine mastitis. Current Issues in Molecular Biology, 44(1), 309-328.

Results

  • The massive amount of data is well presented and a very broad metabolic base were covered. It feels overwhelming though. This is proof that biochemically the prediction of meat characteristics are very difficult and the regulation of the different mechanisms are very complex. Practically it seems nonviable to use these information for breeding selection. Environmental factors would influence expression so much that the genomic knowledge will only be useful on the macro scale. Non the less this research are very interesting and good fundamentally.

Response: We are grateful for all your constructive comments. Our goal was to provide complete information.

Discussion

  • Line 402 - needs reference

Response: We added a reference for this sentence in the revised manuscript.

  • Lines 406 -411 - too long sentence

Response: We applied the mentioned changes in the revised manuscript.

  • The discussion about the gene expression and down and up regulation of physiological properties in the cell is very long and probable good for an science handbook. The practical implementation of this knowledge for animal breeding will be more difficult. Some of the work is part of a literature study and for a scientific paper it should not be added. Maybe it was part of a graduate study and for publication one should be more selective.

Response: We are grateful for all your constructive comments. In this section, we have tried to provide comprehensive information about each gene and its function to inform readers about the function of each gene and regulatory mechanisms. In our previous articles, we used a similar approach and have received positive feedback in this regard. Our strong preference is to present comprehensive information to provide a clear picture of regulatory mechanisms associated with intramuscular fat content in beef cattle. According to the reviewer's comments, information about differentially expressed types of RNAs and their interactions in subnets of the ceRNA regulatory network between eight pairwise comparative groups associated with IMF in beef cattle breeds was presented as Supplementary Material 6 and the discussion became shorter.

> Ghafouri, F., Bahrami, A., Sadeghi, M., Miraei-Ashtiani, S. R., Bakherad, M., Barkema, H. W., & Larose, S. (2021). Omics multi-layers networks provide novel mechanistic and functional insights into fat storage and lipid metabolism in poultry. Frontiers in Genetics, 12, 646297.

> Sadeghi, M., Bahrami, A., Hasankhani, A., Kioumarsi, H., Nouralizadeh, R., Abdulkareem, S. A., ... & Barkema, H. W. (2022). lncRNA–miRNA–mRNA ceRNA Network Involved in Sheep Prolificacy: An Integrated Approach. Genes, 13(8), 1295.

> Naserkheil, M., Ghafouri, F., Zakizadeh, S., Pirany, N., Manzari, Z., Ghorbani, S., ... & Min, K. S. (2022). Multi-omics integration and network analysis reveal potential hub genes and genetic mechanisms regulating bovine mastitis. Current Issues in Molecular Biology, 44(1), 309-328.

> Ghafouri, F., Sadeghi, M., Bahrami, A., Naserkheil, M., Dehghanian Reyhan, V., Javanmard, A., Miraei-Ashtiani, R., Ghahremani, S., Barkema, H.W., Abdollahi-Arpanahi, R. and Kastelic, J.P. (2023). Construction of a circRNA-lincRNA-lncRNA-miRNA-mRNA ceRNA Regulatory Network Identifies Genes and Pathways Linked to Goat Fertility. Frontiers in Genetics, 14, 1195480.

Conclusions

  • Only last sentence was part of conclusion. Interesting concept.

Response: In this section, we have tried to provide a comprehensive summary for the final conclusion regarding the subject of the manuscript.

 Comments on the Quality of English Language: Good

Quality of English Language: Minor editing of English language required

Response: Two co-authors of the manuscript are native English speakers and we have made specific efforts to carefully review and improve the revised manuscript.

Reviewer 2 Report

Reyhan et al. used transcriptome data from five different breeds to build up the ceRNAs network for intramuscular fat contents (IMF) in beef. The authors identified some pathways and candidate genes  (MCU, CYB5R1, and BAG3)  for IMF.  

The strong points of manuscripts include comprehensive selections of data, relevant methods in data/sequence processing, DE analyses, and downstream analyses, excellent presentation of results, and deep discussion on the relevant genes and networks/pathways related to IMF or fat metabolism.

It is not exactly a weakness but as the authors mentioned in lines 708-710, the samples came from different sources and there is some bias to include them in the same analyses. In addition, the DE analyses the authors performed the pairwise comparisons between different breeds. It is not clear which models the authors use for DE analyses, why multiple comparisons were not possible, and did the authors perform normalization of data or removed the low count miRNAs/mRNAs before DE analyses.

Minors:

Some abbreviations in abstract need be defined. 

Table 2: bta-miR-193B, should B or b. The information on the locus and positions is not relevant, I suggest the authors replace them with the group comparisons, abundances (raw counts), fold changes, and FDR.

Figure 4-6: Might increase the font size for the nodes.

In discussion, the authors might not need to list all the genes or multiple genes in the subnets.

Although, the authors listed many miRNAs, lacking discussion about these miRNAs and how they can be used as a biomarker for IFM.

Another aspect is that genes themselves can not be markers, did the authors check if any mutations (SNPs, indels) in these candidate genes have been identified for IMF (in GWAS or functional studies).

Line 715: Why did the authors consider this a novel method, similar approaches have been used in humans and some other species.

Reference style in text might change to Animals journal styles, please check. 

English is fine. 

Author Response

Response to Reviewer 2 Comments

The authors greatly appreciate your insightful comments and have taken them into account to revise the manuscript. Changes are highlighted with yellow text.

 Comments and Suggestions for Authors:

  • Reyhan et al. used transcriptome data from five different breeds to build up the ceRNAs network for intramuscular fat contents (IMF) in beef. The authors identified some pathways and candidate genes (MCU, CYB5R1, and BAG3) for IMF.
  • The strong points of manuscripts include comprehensive selections of data, relevant methods in data/sequence processing, DE analyses, and downstream analyses, excellent presentation of results, and deep discussion on the relevant genes and networks/pathways related to IMF or fat metabolism.
  • It is not exactly a weakness but as the authors mentioned in lines 708-710, the samples came from different sources and there is some bias to include them in the same analyses. In addition, the DE analyses the authors performed the pairwise comparisons between different breeds. It is not clear which models the authors use for DE analyses, why multiple comparisons were not possible, and did the authors perform normalization of data or removed the low count miRNAs/mRNAs before DE analyses.

Response: We are grateful for the reviewer’s constructive comments. Regarding data handling and batch effects, samples had an R2 > 0.8 and there was no need for an adjustment of batch effects. A gene expression network was constructed at β = 12 to ensure that the network followed a scale-free topology (R2 ≥ 0.80). If the scale-free topology fit index for the reference dataset reached values above 0.8 for low powers (< 30), the topology of the network is scale-free and therefore, there are no batch effects (Cline et al., 2007; Kadarmideen et al., 2011; Darzi et al., 2021; Johnston et al., 2021). Consequently, there is no need to remove batch effects in a separate step.

In regards to accession numbers GSE, we tried to select the closest samples in this study, and for preventing any sidles we selected high-quality and matched samples. Also, for integrating different data sets, there are two main approaches: 1) integrating data sets using adjusting algorithms; and 2) analyzing each data set separately and considering common results for particular biological pathways (Najafi et al., 2014). In this study, we used the second approach. We added a brief explanation in the revised manuscript, as follows in the revised manuscript:

"The gene expression network was constructed at β = 12 to ensure that the network followed a scale-free topology (R2 ≥ 0.80). If the scale-free topology fit index for the reference dataset reaches values > 0.8 for low powers (< 30), the topology of the network is considered scale-free and, therefore, there are no batch effects [Cline et al., 2007; Kadarmideen et al., 2011; Darzi et al., 2021; Johnston et al., 2021]."

To confirm, we removed the low-count miRNAs/mRNAs before DE analyses. DESeq2 software, which reports the standard error for each shrunken LFC estimate, obtained from the curvature of the coefficient’s posterior at its maximum. For significance testing, DESeq2 uses a Wald test: the shrunken estimate of LFC is divided by its standard error, resulting in a z-statistic, which is compared to a standard normal distribution. The Wald test allows testing of individual coefficients, or contrasts of coefficients, without the need to fit a reduced model as with the likelihood ratio test, though the likelihood ratio test is also available as an option in DESeq2. The Wald test P values from the subset of genes that pass an independent filtering step, are adjusted for multiple testing using the procedure of Benjamini and Hochberg (Love et al., 2014).

> Cline, M.S., Smoot, M., Cerami, E., Kuchinsky, A., Landys, N., Workman, C., Christmas, R., Avila-Campilo, I., Creech, M., Gross, B., Hanspers, K. (2007). Integration of biological networks and gene expression data using Cytoscape. Nature Protocols, 2(10), 2366-2382.

> Darzi, M., Gorgin, S., Majidzadeh-A, K., Esmaeili, R. (2021). Gene co-expression network analysis reveals immune cell infiltration as a favorable prognostic marker in non-uterine leiomyosarcoma. Scientific Reports, 11(1), 1-11.

> Johnston, D., Earley, B., McCabe, M.S., Kim, J., Taylor, J.F., Lemon, K., Duffy, C., McMenamy, M., Cosby, S.L., Waters, S.M. (2021). Messenger RNA biomarkers of Bovine Respiratory Syncytial Virus infection in the whole blood of dairy calves. Scientific Reports, 11(1), 1-7.

> Kadarmideen, H.N., Watson-Haigh, N.S., Andronicos, N.M. (2011). Systems biology of ovine intestinal parasite resistance: disease gene modules and biomarkers. Molecular Biosystems, 7(1), 235-246.

> Najafi, A., Bidkhori, G., H Bozorgmehr, J., Koch, I., Masoudi-Nejad, A. (2014). Genome scale modeling in systems biology: algorithms and resources. Current Genomics, 15(2), 130-159.

> Love, M. I., Huber, W., & Anders, S. (2014). Moderated estimation of fold change and dispersion for RNA-seq data with DESeq2. Genome biology, 15(12), 1-21.

Minors:

  • Some abbreviations in abstract need be defined.

Response: We applied the mentioned changes in the abstract.

  • Table 2: bta-miR-193B, should B or b. The information on the locus and positions is not relevant, I suggest the authors replace them with the group comparisons, abundances (raw counts), fold changes, and FDR.

Response: According to the reference articles, "b" is used. Also, in order to report comprehensive information about identified miRNAs, in addition to the previous information, we added three columns including "Fold Changes (FC)", "p-Value" and "FDR" to Table 2.

  • Figure 4-6: Might increase the font size for the nodes.

Response: If we increase the font size of the nodes, due to the high number of nodes, they occupy a lot of network space. Therefore in this case, we have tried to provide high-quality Figures so that readers can read the names of the nodes by zooming in on the Figures.

  • In discussion, the authors might not need to list all the genes or multiple genes in the subnets.

Response: We are grateful for all your constructive comments. In this section, we have tried to provide comprehensive information about each gene and its function to inform readers about the function of each gene and regulatory mechanisms. In our previous articles, we used a similar approach and have received positive feedback in this regard. Our strong preference is to present comprehensive information to provide a clear picture of regulatory mechanisms associated with intramuscular fat content in beef cattle. According to the reviewer's comments, information about differentially expressed types of RNAs and their interactions in subnets of the ceRNA regulatory network between eight pairwise comparative groups associated with IMF in beef cattle breeds was presented as Supplementary Material 6 and the discussion became shorter.

> Ghafouri, F., Bahrami, A., Sadeghi, M., Miraei-Ashtiani, S. R., Bakherad, M., Barkema, H. W., & Larose, S. (2021). Omics multi-layers networks provide novel mechanistic and functional insights into fat storage and lipid metabolism in poultry. Frontiers in Genetics, 12, 646297.

> Sadeghi, M., Bahrami, A., Hasankhani, A., Kioumarsi, H., Nouralizadeh, R., Abdulkareem, S. A., ... & Barkema, H. W. (2022). lncRNA–miRNA–mRNA ceRNA Network Involved in Sheep Prolificacy: An Integrated Approach. Genes, 13(8), 1295.

> Naserkheil, M., Ghafouri, F., Zakizadeh, S., Pirany, N., Manzari, Z., Ghorbani, S., ... & Min, K. S. (2022). Multi-omics integration and network analysis reveal potential hub genes and genetic mechanisms regulating bovine mastitis. Current Issues in Molecular Biology, 44(1), 309-328.

> Ghafouri, F., Sadeghi, M., Bahrami, A., Naserkheil, M., Dehghanian Reyhan, V., Javanmard, A., Miraei-Ashtiani, R., Ghahremani, S., Barkema, H.W., Abdollahi-Arpanahi, R. and Kastelic, J.P. (2023). Construction of a circRNA-lincRNA-lncRNA-miRNA-mRNA ceRNA Regulatory Network Identifies Genes and Pathways Linked to Goat Fertility. Frontiers in Genetics, 14, 1195480.

  • Although, the authors listed many miRNAs, lacking discussion about these miRNAs and how they can be used as a biomarker for IFM.

Response: The general role of miRNAs has been mentioned in the introduction of the manuscript, and according to most of the articles published up to now, as well as the incompleteness of the scientific information on different types of non-coding RNAs, we have tried to provide some information extent indirectly with regard to the gene's function and its regulatory role. Surely, in the articles that will be published in the future, with the increase of information related to each of the ncRNAs, studies with higher precision and accuracy can be reported in this regard. Therefore, in this manuscript, we have not gone too much into the details regarding the function of ncRNAs especially miRNAs, and have discussed more about the function of their target genes.

  • Another aspect is that genes themselves can not be markers, did the authors check if any mutations (SNPs, indels) in these candidate genes have been identified for IMF (in GWAS or functional studies)?

Response: In this study, our main focus has been on one of the layers of Omics technology, i.e., transcriptomics, and we have tried to identify genes and ncRNAs with expression differences (common and non-common) between different breeds. Certainly, studies have been reported in relation to SNPs and indels, and we intend to examine the genomic data in future studies and integrate them with the outputs of this manuscript in order to provide more reliable and practical results that can be used in breeding strategies.

  • Line 715: Why did the authors consider this a novel method, similar approaches have been used in humans and some other species?

Response: We applied some changes in the sentence. It can be mentioned that in the field of animal sciences, many studies that have considered several different breeds at the same time and have used ceRNA regulatory networks in this regard have not been reported so far.

  • Reference style in text might change to Animals journal styles, please check.

Response: We applied changes in the references (in the text and references section) in regard to Animals journal styles.

Comments on the Quality of English Language: English is fine.

Quality of English Language: Minor editing of English language required

Response: Two co-authors of the manuscript are native English speakers and we have made specific efforts to carefully review and improve the revised manuscript.